# JUST-IN-TIME SECURITY PATCH DETECTION - LLM AT THE RESCUE FOR DATA AUGMENTATION

## ABSTRACT

In the face of growing vulnerabilities found in open-source software, the need to identify discreet security patches has become paramount. The lack of consistency in how software providers handle maintenance often leads to the release of security patches without comprehensive advisories, leaving users vulnerable to unaddressed security risks. To address this pressing issue, we introduce a novel security patch detection system, LLMDA, which capitalizes on Large Language Models (LLMs) and code-text alignment methodologies for patch review, data enhancement, and feature combination. Within LLMDA, we initially utilize LLMs for examining patches and expanding data of PatchDB and SPI-DB, two security patch datasets from recent literature. We then use labeled instructions to direct our LLMDA, differentiating patches based on security relevance. Following this, we apply a PTFormer to merge patches with code, formulating hybrid attributes that encompass both the innate details and the interconnections between the patches and the code. This distinctive combination method allows our system to capture more insights from the combined context of patches and code, hence improving detection precision. Finally, we devise a probabilistic batch contrastive learning mechanism within batches to augment the capability of the our LLMDA in discerning security patches. The results reveal that LLMDA significantly surpasses the start of the art techniques in detecting security patches, underscoring its promise in fortifying software maintenance.

## 1 INTRODUCTION

The widespread adoption of open-source software (OSS) has been a transformative force in software development. OSS projects have become cornerstones of modern computing infrastructure, driving innovation and fostering a culture of collaboration. However, as with all technologies, the broad use of OSS is accompanied by its own challenges. As indicated in the 2021 Open Source Security and Risk Analysis (OSSRA) report (Synopsys, 2021), the rapid expansion of OSS has led to a corresponding surge in vulnerabilities. These vulnerabilities, when exploited, enable attackers to perform "N-day" attacks against unpatched software systems. These attacks often have severe consequences. In November 2022, Threat Analysis Group discovered zero-day exploit chains targeting Android and iOS. These exploits were delivered via SMS bit.ly links to users in Italy, Malaysia, and Kazakhstan, related to CVE-2022-42856[1] and CVE-2022-4135[2]. Clicking the links redirected users to exploit-hosting pages for Android or iOS, followed by redirects to seemingly legitimate sites, including an Italian logistics company's shipment tracking page and a popular Malaysian news website. Such scenarios underline the critical importance of a swift and effective response to identified vulnerabilities in OSS.

One of the primary defenses against the attacks to the OSS is timely software patching. Patches are code updates that address identified vulnerabilities, fix performance issues, or add new features to an existing software system (Li & Paxson, 2017; Tan et al., 2021). However, the ever-increasing number of submitted patches can overwhelm reviewers and system administrators. Additionally, the complexity of the patch management process, which includes the collection, testing, validation, and scheduling, can often result in delays in software updates (Dissanayake et al., 2022).

---

[1]https://nvd.nist.gov/vuln/detail/cve-2022-42856
[2]https://nvd.nist.gov/vuln/detail/cve-2022-4135

In order to safeguard the system against potential attacks and streamline the management of the overall process, it is imperative that we place a strong emphasis on prioritizing critical patches, particularly those pertaining to important security matters. Many researchers (Wang et al., 2023a) invest significant efforts to address such challenges. For example, there are approaches leveraging machine learning algorithms with syntax features (Wang et al., 2020; 2019; Tian et al., 2012) or sequential deep neural networks to handle the patches as sequential data (Zhou et al., 2021; Wang et al., 2021b). Claiming that such methods lack program semantics and produce a high false-positive rate, a state-of-the-art approach, GraphSPD (Wang et al., 2023a) employs the graph structure of the source code to detect the security patches.

While the state-of-the-art technique proposed by Wang et al. (2023a) successfully captures context within patches and outperforms other existing techniques, it is worth noting that the approach focuses on local code segments, which may not capture the broader context of how functions or modules interact. In the current wave of Large Language Models (LLMs), recent studies (Li et al., 2023; Sun et al., 2023; Su & McMillan, 2023) have shown that LLMs can adeptly capture the essential context and tokens within source code. Furthermore, other studies (Wei et al., 2021; Chung et al., 2022; Dai et al., 2023) have revealed that the incorporation of natural language instructions using appropriate templates significantly improves learning performance. These indications suggest that embarking on a language-centric approach would be worthwhile.

In this paper, we propose an LLM-powered model that aligns multi-modal input for more accurate security patch detection. Given a patch that already includes a source code and the description (description given by the commit message), we leverage an LLM to generate its explanation and design an instruction to better guide our model toward the target task. Inspired by the current success of text generation of LLaMa (Touvron et al., 2023), we employ the inference power of the LLaMa 7b model while we follow the same procedure of the prior works (Wei et al., 2021; Chung et al., 2022) to design the instructions. Feeding four different input modalities, we utilize a current state-of-the-art code embedding model, CodeT5+ (Wang et al., 2023b), and LLaMa 7b to generate each embedding. These embeddings are then fed into LLMDA which has been constructed to align the different embeddings to a single space while considering the characteristics of the inputs. Inside LLMDA, the instructions play a crucial role as they allow the model to be label-wise learning. Once all the embeddings are aligned, we take advantage of the Stochastic Contrastive Learning module (Oh et al., 2018) to get the final binary output which is either security or non-security.

The evaluation focused on the proficiency of the framework in detecting security patches and the efficacy of the principal design decisions. Based on a preliminary literature review, we carefully selected our baseline methods and evaluation metrics. The experimental results show that our framework consistently outperforms the baseline methods (i.e., TwinRNN (Wang et al., 2021b) and GraphSPD (Wang et al., 2023a)) on both of our target datasets (i.e., PatchDB (Wang et al., 2021a) and SPI-DB (Zhou et al., 2021)). Specifically, LLMDA achieves 42.86% and 20.05% better performance than the state-of-the-art on both of our target datasets, respectively. We also emphasize the practical applicability of our framework by validating its performance in terms of detection precision.

Our contributions are as follows:

- We introduce an innovative security patch detection framework, LLMDA, leveraging LLMs for patch analysis and data augmentation, while aligning various modalities. This enables our system to extract richer information from the joint context of patches and code, boosting detection accuracy.
- We illustrate that a language-centric approach, coupled with a well-designed framework, can yield significant performance improvements in the context of security patch detection.
- Our results underline the effectiveness of our approach and its potential for real-world applications by showcasing the precise detection capability in secure software maintenance.

## 2 METHODOLOGY

In our proposed methodology, the central aim is to unify the embeddings of patches and textual descriptions, align them in a shared embedding space, and efficiently classify patches as security or non-security. As shown in Figure 1, we have four different inputs: Patch, Explanation, Description,

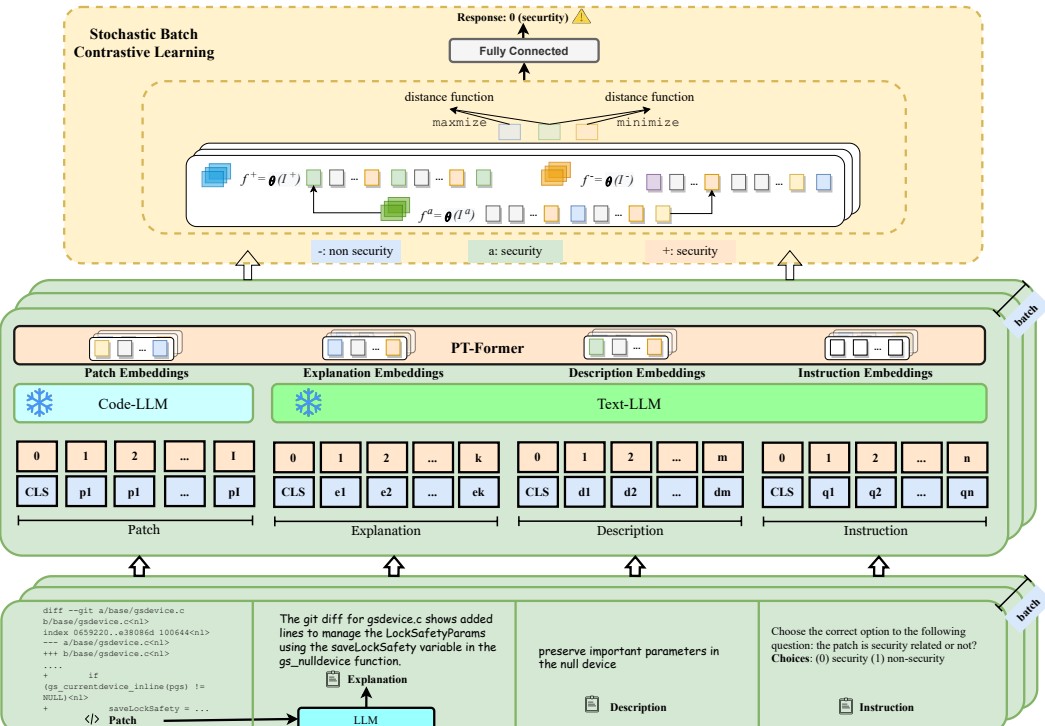

**Figure 1:** Overview of our model and a practical example.

and Instruction (We explain the input in Sec A.1). We use Code-LLM and Text-LLM to obtain their embeddings. Furthermore, we feed four multimodal embeddings into our designed PT-Former to align and fuse the embeddings. Finally, we design a stochastic batch contrastive learning to learn the difference between security and non-security data points inside the given mini-batch.

### 2.1 EMBEDDING GENERATION

**Patch Embeddings with CodeT5+**: Consider $P$ as a matrix representation of a code patch where each row corresponds to a token's representation. The transformation function $f_{\text{CodeT5+}}$ applied on $P$ yields an embedding $E_p$:

$$E_p = f_{\text{CodeT5+}}(P) = \mathcal{F}(P \cdot W_p + b_p) \tag{1}$$

where $W_p$ is a weight matrix, $b_p$ is a bias vector, and $\mathcal{F}$ denotes a non-linear activation function. The matrix multiplication and subsequent activation capture the intricate relationships between different tokens of the patch.

**Textual Embeddings with LLaMa (7b)**: Let $T$ be a matrix representation of a text where each row corresponds to a word's representation. The transformation function $f_{\text{LLaMa}}$ applied on $T$ produces an embedding $E_t$:

$$E_t = f_{\text{LLaMa}}(T) = \mathcal{G}(T \cdot W_t + b_t) \tag{2}$$

where $W_t$ is a weight matrix for the textual transformation, $b_t$ is the corresponding bias vector, and $\mathcal{G}$ is another non-linear activation function.

In our study, we have three text-like inputs: instruction, explanation, and description. Then we get $E_t = E_e \oplus E_d \oplus E_i$, where $\oplus$ means concatenation operation.

## 2.2 EMBEDDING ALIGNMENT WITH PT-FORMER

As shown in Figure 2, the Patch-Text Aligner (PT-Former) serves as a nexus between code- and text-based embeddings.

**Hierarchical Attention Mechanisms** Attention mechanisms involve computing weighted combinations of input vectors. For the self-attention mechanism, considering multi-head attention with $h$ heads:

$$W_{Q_i}, W_{K_i}, W_{V_i} \sim \mathcal{N}(0, 1), \quad i = 1, ..., h \tag{3}$$

Given the above weight matrices for query, key, and value, respectively, our self-attention mechanism over the explanations $E_{\text{expl}}$ is computed as:

$$\mathcal{AH}_i(E_{\text{expl}}) = \text{Softmax}\left(\frac{E_{\text{expl}}W_{Q_i}(E_{\text{expl}}W_{K_i})^T}{\sqrt{d}}\right) E_{\text{expl}}W_{V_i} \tag{4}$$

where $d$ is the dimensionality of the embeddings. After obtaining attention outputs from all heads, we concatenate them:

$$S_{\text{expl}} = \text{Concat}(\mathcal{AH}_1, ..., \mathcal{AH}_h)W_O \tag{5}$$

where $W_O$ is the output weight matrix.

The cross-attention with patch embeddings $E_p$ follows a similar formulation: $C_{\text{expl}} = \text{Softmax}\left(\frac{S_{\text{expl}}W_Q(E_pW_K)^T}{\sqrt{d}}\right) S_{\text{expl}}W_V$

**Embedding Fusion and Non-linear Transformation** With the processed embeddings at hand, we then expose them to feedforward layers. Each feedforward layer is composed of two dense layers with a ReLU activation in between: $F(x) = W_2 \cdot \text{ReLU}(W_1 \cdot x + b_1) + b_2$. Given this function, our transformation of the embeddings can be written as: $E_{\text{expl}} = F(C_{\text{expl}})$, $E_{\text{desc}} = F(S_{\text{desc}})$, $E_{\text{instr}} = F(S_{\text{instr}})$. Then we obtain the final representation of multi-modal embedding $\mathcal{O} = [E_p, E_{\text{expl}}, E_{\text{desc}}, E_{\text{instr}}]$

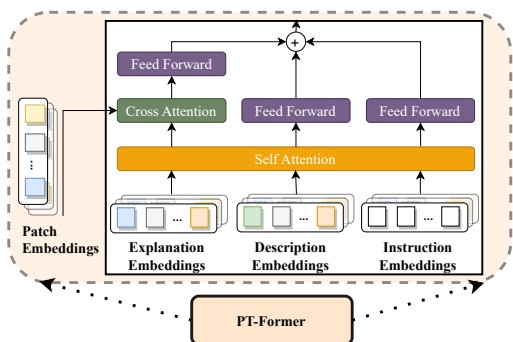

**Figure 2:** Model architecture of PT-Former.

**Label-wise Instruction Incorporation** Expired by the paradigms in InstructionBLIP (Dai et al., 2023), instruction with questions and labels inside can provide two advantages: One is to provide guidance to train models in the direction of answering the security question; As we involve labels inside the instruction, it provides the possibility of building a relationship between inputs and labels with the calculation of their high-dimensional embeddings, we leverage instruction in a label-wise manner. In conclusion, Instruction guides the model to focus on particular aspects of the data, thereby improving the representational efficiency for the targeted downstream task.

## 2.3 PROBABILISTIC BATCH EMBEDDING CALCULATION (PBCL)

**Batch Probabilistic Input Embedding** For a batch of multi-modal inputs $\mathcal{B}$, consisting of multiple multi-modal inputs $\mathcal{O}_i$, clips are sampled and denoted as $c_{n,i}$ for the $i$th input ($i \leqq 4$).

Given an input $\mathcal{O}_i$, let $\{c_{1,i}, \ldots, c_{N,i}\}$ be a set of clips from $\mathcal{O}_i$. The output of the backbone network parameterized by $\theta$ for each clip is: $v_{c_{n,i}} = f_\theta(c_{n,i})$.

The probability distribution for each clip is then given by:

$$p(z \mid c_{n,i}) \sim \mathcal{N}\left(g_\mu\left(v_{c_{n,i}}\right), \text{diag}\left(g_\sigma\left(v_{c_{n,i}}\right)\right)\right) \tag{6}$$

where $g_\mu$ is a fully connected ($FC$) layer followed by LayerNorm and $\ell_2$ normalization, and $g_\sigma$ is a separate $FC$ layer. The $i$-th multi-modal input in the batch can be represented as:

$$p(z \mid \mathcal{V}_i) \sim \sum_{n=1}^{N} \mathcal{N}\left(z; g_\mu\left(v_{c_{n,i}}\right), \mathrm{diag}\left(g_\sigma\left(v_{c_{n,i}}\right)\right)\right) \tag{7}$$

From each distribution, $p(z \mid \mathcal{V}_i)$, $K$ embeddings are sampled. Using the reparameterization trick for stable training:

$$z_i^{(k)} = \sigma(\mathcal{V}_i) \cdot \epsilon^{(k)} + \mu(\mathcal{V}_i) \tag{8}$$

where $\mu(\mathcal{V}_i), \sigma(\mathcal{V}_i)$ are the mean and the standard deviation of $p(z \mid \mathcal{V}_i)$, and the $\epsilon^{(k)}$ values are sampled from a $D$ dimensional unit Gaussian distribution.

**Batch Mining of Positive and Negative Pairs** Consider a batch $\mathcal{B}$ of size $B$. Within this batch, each multi-modal input $\mathcal{V}_i$ has multiple probabilistic embeddings $\{z_i^{(1)}, z_i^{(2)}, \dots, z_i^{(K)}\}$ due to the stochastic nature of our model.

We aim to find informative positive and negative pairs within this batch. The Bhattacharyya distance for any two probabilistic embeddings, sampled from the multi-modal inputs $i$ and $j$ in batch $\mathcal{B}$, is defined as:

$$\mathrm{dist}_\mathcal{B}\left(z_i^{(k)}, z_j^{(k')}\right) = \frac{1}{4}\left(\log\left(\frac{1}{4}\left(\frac{\sigma_i^2}{\sigma_j^2} + \frac{\sigma_j^2}{\sigma_i^2} + 2\right)\right) + \lambda \cdot \frac{\left(z_i^{(k)} - z_j^{(k')}\right)^\top \left(z_i^{(k)} - z_j^{(k')}\right)}{\sigma_i^2 + \sigma_j^2}\right) \tag{9}$$

For each pair of multi-modal inputs in the batch, we estimate the average distance over all their probabilistic embeddings as:

$$\mathrm{avg\_dist}_\mathcal{B}\left(\mathcal{V}_i, \mathcal{V}_j\right) = \frac{1}{K^2}\sum_{k=1}^{K}\sum_{k'=1}^{K}\mathrm{dist}_\mathcal{B}\left(z_i^{(k)}, z_j^{(k')}\right) \tag{10}$$

Then, within this batch, positive pairs are defined using a distance threshold $\tau$:

$$\mathcal{P}_\mathcal{B} = \{(\mathcal{V}_i, \mathcal{V}_j) \in \mathcal{B} \mid \mathrm{avg\_dist}_\mathcal{B}\left(\mathcal{V}_i, \mathcal{V}_j\right) < \tau \text{ or } i = j\} \tag{11}$$

Negative pairs within the batch are defined as: $\mathcal{N}_\mathcal{B} = \mathcal{B} \setminus \mathcal{P}_\mathcal{B}$. With this batch-aware formulation, we explicitly consider the relationship between multi-modal inputs and their probabilistic embeddings within a batch, ensuring a more targeted and efficient training approach.

**Stochastic Batch Contrastive Loss (SBCL)**

To effectively utilize the positive and negative pairs mined from the batch, we introduce the Batch Stochastic Contrastive Loss. This loss aims to bring together the embeddings of positive pairs and push apart the embeddings of negative pairs, making full use of the stochastic nature of our model.

For each pair of multi-modal inputs $(\mathcal{V}_i, \mathcal{V}_j)$ within the batch, their loss contribution based on their average distance $\mathrm{avg\_dist}_\mathcal{B}$ is:

$$
\begin{aligned}
L_{\mathrm{pair}}(\mathcal{V}_i, \mathcal{V}_j) = &-\log\left(\frac{\exp(-\mathrm{avg\_dist}_\mathcal{B}(\mathcal{V}_i, \mathcal{V}_j)/\tau)}{\sum_{\mathcal{V}_k \neq \mathcal{V}_i} \exp(-\mathrm{avg\_dist}_\mathcal{B}(\mathcal{V}_i, \mathcal{V}_k)/\tau)}\right) \\
&\text{s.t. } (\mathcal{V}_i, \mathcal{V}_j) \in \mathcal{P}_\mathcal{B}, \quad or \\
&-\log\left(1 - \frac{\exp(-\mathrm{avg\_dist}_\mathcal{B}(\mathcal{V}_i, \mathcal{V}_j)/\tau)}{\sum_{\mathcal{V}_k \neq \mathcal{V}_i} \exp(-\mathrm{avg\_dist}_\mathcal{B}(\mathcal{V}_i, \mathcal{V}_k)/\tau)}\right) \\
&\text{s.t. } (\mathcal{V}_i, \mathcal{V}_j) \notin \mathcal{P}_\mathcal{B}
\end{aligned}
\tag{12}
$$

Thus, the total batch stochastic contrast loss for the entire batch is:

$$L_{\text{contrastive}} = \frac{1}{B(B-1)} \sum_{\mathcal{V}_i, \mathcal{V}_j \in \mathcal{B}, i \neq j} L_{\text{pair}}(\mathcal{V}_i, \mathcal{V}_j) \tag{13}$$

where $B$ refers to the size of the batch.

Minimizing $L_{\text{contrastive}}$ encourages the model to produce embeddings that are closer for multi-modal inputs in the positive pair set and further apart for multi-modal inputs in the negative pair set. This alignment with the underlying structure of the batched data ensures efficient and effective representation learning.

### 2.4 Prediction and Training Layer for Security Patch Detection

Given that we have multi-modal input embeddings from the previous stages, we aim to perform binary detection on multi-modal inputs. Let's use $e_{\mathcal{V}_i}$ to represent the embedding of the $i$-th multi-modal input in the batch $\mathcal{B}$.

**Prediction Layer** The prediction layer will consist of a series of operations that transform our embeddings into a probability space suitable for binary classification. Given the batched nature of our processing, let's represent this operation in matrix form:

For a batch $\mathcal{B}$, we can represent the embeddings in the matrix form $E \in \mathbb{R}^{B \times D}$, where $B$ is the batch size and $D$ is the dimension of the embeddings. The probability predictions for the batch can be computed as: $P = \sigma(EW + \mathbf{b})$ where $W \in \mathbb{R}^{D \times 1}$ is the weight matrix; $\mathbf{b} \in \mathbb{R}^{B \times 1}$ is the bias vector, replicated for each instance in the batch; $\sigma$ is the sigmoid function applied element-wise.

**Training Layer** For binary classification, the Binary Cross-Entropy (BCE) loss is commonly employed. For our batched embeddings and predictions, the BCE loss is given by:

$$L_{\text{BCE}} = -\frac{1}{B} \sum_{i=1}^{B} (y_i \log(p_i) + (1 - y_i) \log(1 - p_i)) \tag{14}$$

where $y_i$ represents the true binary label of the $i$-th multi-modal input in the batch. $p_i$ represents the predicted probability for the $i$-th multi-modal input.

In an end-to-end training regime, both the contrastive loss from the previous sections and the BCE loss are combined: $L_{\text{total}} = L_{\text{contrastive}} + L_{\text{BCE}}$.

**Optimization** The combined loss $L_{\text{total}}$ is minimized using gradient-based optimization techniques. The gradient of $L_{\text{total}}$ w.r.t. the network parameters are computed using backpropagation. Optimizers like Adam or SGD can then be employed to iteratively refine the model parameters for optimal performance.

## 3 Experimental Setup

This section outlines our evaluation metrics, compares our approach to established methods, and presents the core research inquiries.

### 3.1 Dataset

The study uses two primary datasets for experiments: PatchDB, which comprises 36K patches from open-source projects like Linux and MySQL, and SPI-DB, focusing on patches from projects such as Linux and FFmpeg, of which only 25,790 patches are publicly available. These datasets offer a diverse array of patch types for comprehensive security patch detection assessment. The research further employs the latest LLMs such as ChatGPT, GPT-4, and others for synthetic data generation, especially using models like dolly-v2-12b and StableVicuna13B. Through these LLMs, the study seeks to generate explanations for patches by feeding specific prompts, ultimately producing a dataset in the format of <patch, explanation, description, instruction>. More details are discussed in Section A.1 in Appendix.

## 3.2 EVALUATION METRICS

**+Recall** and **-Recall**, as detailed in Tian et al. (2022), are tailored metrics for gauging patch correctness. The former measures proficiency in predicting accurate patches, whereas the latter evaluates capability in excluding erroneous ones.

**AUC and F1-score**. To discern patch accuracy, we devised an NLP-based deep learning classifier. Our methodology's effectiveness is gauged using the renowned AUC and F1-score metrics. The F1-score, being the harmonic mean of precision and recall, is specifically tailored for pinpointing correct patches (Hossin & Sulaiman, 2015).

**True-Positive Rate (TPR)**. Also termed as sensitivity, TPR captures the classifier's prowess in spotting positive cases. For security patches, it denotes the fraction of legitimate patches accurately identified, with a superior TPR revealing minimal missed security patches and thereby curtailing vulnerabilities (Wang et al., 2023a).

## 3.3 BASELINE METHODS

**TwinRNN**: Highlighted in the state-of-the-art (Wang et al., 2023a) and building upon insights from literature works (Zhou et al., 2021; Wang et al., 2021b), TwinRNN boasts a distinct architecture grounded on RNN solutions for pinpointing security patches. The "twin" nomenclature stems from its bifurcated RNN module design, where each segment handles either pre-patch or post-patch code sequences.

**GraphSPD**: In the spectrum of patch detection, TwinRNN is a notable contender. However, GraphSPD introduces a divergent approach and execution, outperforming others.

## 3.4 INVESTIGATIVE QUESTIONS

**RQ-1** How effective is LLMDA at detecting security patches?
**RQ-2** How do principal design decisions influence LLMDA's performance?

## 4 EXPERIMENT RESULTS

### 4.1 [RQ-1:] OVERALL PERFORMANCE

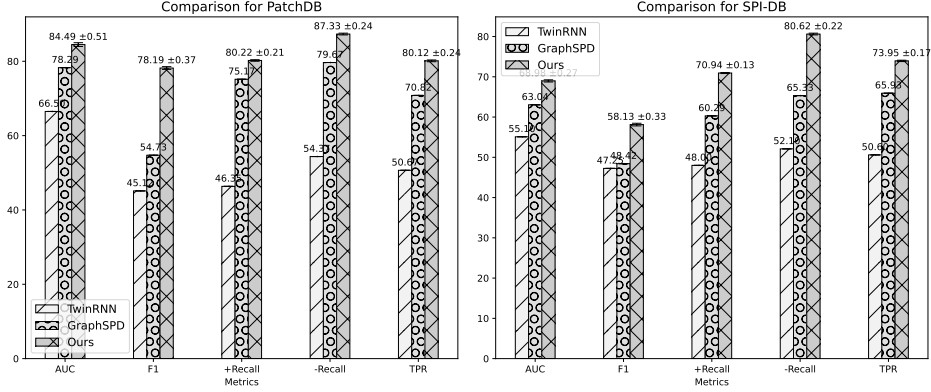

**Figure 3:** Comparison of TwinRNN, GraphSPD, and our LLMDA on PatchDB and SPI-DB with various metrics (%).

**Performance Overview**   As illustrated in Figure 3, our model, **LLMDA**, consistently surpasses its competitors on both the PatchDB and SPI-DB datasets. On PatchDB, LLMDA showcases an AUC of 84.49%, paired with a significant F1-score of 78.19%. For SPI-DB, it posts an AUC of 68.98% and an F1-score of 58.13%. However, it is crucial to stress that equating performances on PatchDB and SPI-DB can be deceptive due to their inherent data peculiarities. Both datasets essentially serve to spotlight the edge of our solution over existing methods.

**Benchmarks Face-Off** In our rigorous comparison with renowned models GraphSPD and Twin-RNN, founded on uniform training and test divisions, Figure 3 captures the essence.

**Dominance Demonstrated.** On PatchDB, LLMDA stands tall against TwinRNN with superior AUC metrics and a pronounced enhancement in the F1-score. Even when pitched against Graph-SPD, LLMDA's prowess is unmistakable in both AUC and F1-score areas. Turning our gaze to SPI-DB, LLMDA trumps both TwinRNN and GraphSPD. An accentuated AUC emphasizes LLMDA's adeptness in class differentiation, and a bolstered F1-score speaks of its balanced output. Pertinently, surges in Recall+ and TPR reflect the model's finesse in spot-on positive instance identification, while better Recall- stats underscore adept negative instance discernment.

**Real-world Relevance.** Precision and the false positive rate are pivotal in ensuring streamlined updates and productivity enhancements. As Figure 3 conveys, LLMDA validates that a notable majority of the projected security patches align with security principles. Regarding SPI-DB, the precision of LLMDA stands out, and its reduced false positive rate distinguishes it as an excellent choice for real-world scenarios.

> **RQ-1 Insights:** *Figure 3 affirms* **LLMDA***'s supremacy over benchmarks on both PatchDB and SPI-DB. Models like GraphSPD and TwinRNN, exhibiting commendable precision,* **LLMDA** *emerges as the exemplar in security patch detection, perfectly suited for real-world challenges.*

## 4.2 [RQ-2:] ABLATION STUDY

**Efficiency of different input** To discern the relative importance of the different levels of context used in our approach — specifically patch (**PT**), explanation (**EX**), description (**DP**), and instruction (**IS**) — we embarked on an ablation study. By systematically omitting one of these inputs at a time, we generated three variants of LLMDA: LLMDA $_{PT-}$ (without patch), LLMDA $_{EX-}$ (sans explanation), LLMDA $_{DP-}$ (devoid of DP), and LLMDA $_{IS-}$ (devoid of IS). The goal of this study was to shed light on how each contextual level contributes to the overall performance of security patch detection.

**Table 1:** Comparison of LLMDA inputs with and without PBCL on PatchDB and SPI-DB with various metrics (%)

| Method | | With PBCL | | | | | | Without PBCL | | | | |
|---|---|---|---|---|---|---|---|---|---|---|---|---|
| | Dataset | AUC | F1 | +Recall | -Recall | TPR | Dataset | AUC | F1 | +Recall | -Recall | TPR |
| LLMDA | PatchDB | 84.49 | 78.19 | 80.22 | 87.33 | 80.12 | PatchDB | 82.93 | 76.45 | 78.72 | 85.81 | 78.60 |
| | SPI-DB | 68.98 | 58.13 | 70.94 | 80.62 | 73.95 | SPI-DB | 67.43 | 56.61 | 69.45 | 79.10 | 72.91 |
| LLMDA $_{PT-}$ | PatchDB | 83.77 | 77.28 | 79.72 | 86.54 | 79.76 | PatchDB | 81.22 | 75.34 | 77.75 | 85.10 | 77.29 |
| | SPI-DB | 68.47 | 57.82 | 70.49 | 80.30 | 73.75 | SPI-DB | 66.02 | 55.91 | 67.34 | 78.80 | 71.57 |
| LLMDA $_{EX-}$ | PatchDB | 83.24 | 76.73 | 79.01 | 86.09 | 79.39 | PatchDB | 81.78 | 74.33 | 77.61 | 84.11 | 77.56 |
| | SPI-DB | 68.27 | 57.57 | 70.23 | 80.07 | 73.50 | SPI-DB | 66.09 | 56.73 | 68.13 | 78.58 | 71.31 |
| LLMDA $_{DP-}$ | PatchDB | 77.99 | 71.45 | 74.47 | 81.77 | 74.58 | PatchDB | 76.55 | 68.99 | 73.85 | 79.32 | 74.16 |
| | SPI-DB | 63.43 | 52.66 | 66.46 | 76.17 | 69.48 | SPI-DB | 61.87 | 51.76 | 65.33 | 74.65 | 68.79 |
| LLMDA $_{IS-}$ | PatchDB | 82.51 | 76.14 | 78.55 | 85.64 | 79.02 | PatchDB | 80.62 | 74.51 | 77.85 | 84.44 | 78.17 |
| | SPI-DB | 67.93 | 57.25 | 69.90 | 79.62 | 73.27 | SPI-DB | 66.23 | 55.51 | 68.57 | 78.16 | 71.78 |

**Importance of PBCL** The introduction of PBCL (Security-Based Contextual Learning) into the LLMDA model was posited as a key driver for enhancing the performance metrics associated with security patch detection. To validate this assertion, an ablation study was performed, isolating the impact of PBCL by evaluating LLMDA with and without its integration.

Table 1 provides a comprehensive overview of the results of this study. The differential in performance across all methods and datasets—PatchDB and SPI-DB—emphasizes the significance of PBCL. Metrics like AUC, F1-score, +Recall, -Recall, and TPR, consistently showed an uptick ranging from 1.00 to 2.50 percentage points when PBCL was incorporated. The degradation in the performance of the Without PBCL variant reaffirms PBCL's role in adding a pivotal layer of contextual sophistication. By virtue of providing an enriched understanding of the patches, PBCL evidently augments the robustness of LLMDA's detection capabilities. This study underscores PBCL's role as an indispensable component in the overarching architecture, optimizing the model for more precise and nuanced security patch detection.

**Contextual Level Contributions** Building upon the foundation set by the integration of PBCL, LLMDA harnesses various levels of context—patch (**PT**), explanation (**EX**), description (**DP**), and instruction (**IS**). Each of these levels contributes distinctively to the model's efficacy. By intentionally omitting one input at a time, the relative importance of these levels was assessed.

The ablation results reveal a multifaceted interplay of these inputs. The removal of any single context invariably led to a decline in performance, underscoring their collective synergy. However, certain contexts such as the description (LLMDA $_{DP-}$) emerged as particularly influential, as evidenced by pronounced drops in metrics when they were excluded. This indicates the criticality of grasping the overarching narrative or intent behind a patch. In essence, while each context contributes uniquely, they cohesively blend to afford LLMDA its high precision in security patch detection.

> **RQ-2 Insights:** *The clear difference made by leaving out each part of the context, along with the big improvement in performance thanks to PBCL, shows that all these parts are very important together. This exploration not only sheds light on the importance of nuanced multi-level contextual understanding but also underscores the crucial role of PBCL in bolstering the efficacy of* LLMDA.

## 5 RELATED WORK

**Security Patch Analysis: Techniques and Advancements** Patch analysis has witnessed considerable evolution, with a transition from empirical studies and traditional techniques to advanced machine learning methods. Early works such as the empirical study by Li & Paxson (2017) highlighted key behaviors in security patches. Initial tools, like VCCFinder (Perl et al., 2015), utilized traditional machine learning techniques such as SVMs. Rule-driven approaches by Wu et al. (2020) and Huang et al. (2019) targeted common security patches. The integration of automation was exemplified by Soto et al. (2016) and Vulmet (Xu et al., 2020). A significant shift towards machine learning was marked by the Random Forest-based work of Wang et al. (2020). Deep learning, especially through RNNs, emerged as a robust tool in patch identification, as shown by PatchRNN (Wang et al., 2021b) and SPI (Zhou et al., 2021). GraphSPD (Wang et al., 2023a) introduced an innovative graph-based approach to the field.

**Binary Patch Analysis and Techniques** Beyond just security patches, binary patch analysis encompasses differentiation, verification, recognition, and automation. This field has produced notable works such as differentiation methods by Ming et al. (2017) and Duan et al. (2020). Verification tools were introduced by Dai et al. (2020) and Zhang et al. (2021). Recognition of patches was discussed by Xu et al. (2017), and automation in this domain was advanced by works like Duan et al. (2019).

**Deep Learning in Sequential Data and Vulnerability Detection** The application of deep learning has been widespread, especially in handling sequential data and vulnerability detection. Compression techniques for sequential data were introduced by Luo et al. (2021). Vulnerability detection, moving from traditional fuzzing-based techniques such as IoTFuzzer (Chen et al., 2018), has embraced deep learning. Russell et al. (2018) and SySeVR (Li et al., 2021) are noteworthy in this transformation.

## 6 CONCLUSION

In this paper, we proposed a security patch detection framework, named LLMDA, by leveraging the powerful capacity of existing LLMs and aligning various modalities. We evaluated our framework against the state-of-the-art on the most well-known datasets to validate the superiority of our approach design. We found that our approach's design not only outperforms existing methods but also demonstrates a practical level of applicability. Our study revealed that a language-centric approach may hold greater potential than focusing on the characteristics of the source code. Additionally, our ablation study showcases the importance of each module of our framework. In particular, it demonstrates that the description-level context, which encompasses a holistic understanding, has the most significant impact on performance. Lastly, we provide a package to reproduce our experiments which is available at the following address: `https://anonymous.4open.science/status/LLMDA-3AC8`

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

## A APPENDIX

### A.1 DATA AUGMENTATION

This section outlines the datasets utilized in our experiments and provides detailed instructions on the setup.

#### A.1.1 DATASET

**PatchDB** Wang et al. (2021a) presents an extensive array of patches written in C/C++, consisting of 12K security-focused and 24K general patches. This dataset combines patches derived from NVD reference links and direct GitHub commits from 311 notable open-source projects, including the Linux kernel, MySQL, and OpenSSL. The broad range present in this dataset aids in thoroughly assessing the efficacy of security patch detection across a myriad of projects.

**SPI-DB** Zhou et al. (2021) concentrates on patches from projects such as Linux, FFmpeg, Wireshark, and QEMU. However, only the information related to FFmpeg and QEMU, amassing 25,790 patches (10K security and 15K non-security), is publicly accessible. Collectively, these datasets provide a vast variety of patch types, facilitating both cross-project and within-project assessments. They also maintain a balanced representation, addressing both real-world relevance and effective model training.

#### A.1.2 LLMS FOR DATA GENERATION

Newer LLMs, like ChatGPT, GPT-4, Dolly-v2, and StableVicuna, are tailored to handle complex, broad-spectrum tasks. They've shown proficiency in adhering to our data directives. Especially, GPT-4 and ChatGPT have a knack for crafting examples in code-similar languages Tian et al. (2023).

Our study on synthetic data creation leverages these LLMs, ensuring equilibrium between open and proprietary models. Specifically, we engage dolly-v2-12b, an offshoot of EleutherAI's Pythia-12b Biderman et al. (2023), honed using roughly 15K directives from Databricks experts. We also utilize StableVicuna13B, an RLHF-optimized Vicuna iteration suited for diverse conversational and guideline-driven datasets. Notably, Vicuna is a publicly available LLaMA variant Touvron et al. (2023). In our study, we fed several prompts into LLMs to explain the patches. However, we only discuss one of them in our main content and keep others in appendix section. The prompt discussed here is "Could you provide a concise summary of the specified patch?". We give an example shown in Figure 1 to show the ability of chatGPT for generating explanation for a given patch. Since our tasks are binary-detection, we can make input text aware the existence of vectors of labels. Thus, we match a instruction "Choose the coorrect option to the following question: the patch is security related or not? Choices: (0) security (1) non-security" for all items in the dataset. Finally, we have the dataset like **<patch, explanation, description, instruction >**.

## A.2 VULNERABILITY TYPE AND LLMDA'S PERFORMANCE IN DIFFERENT PATCH CLASSIFICATIONS

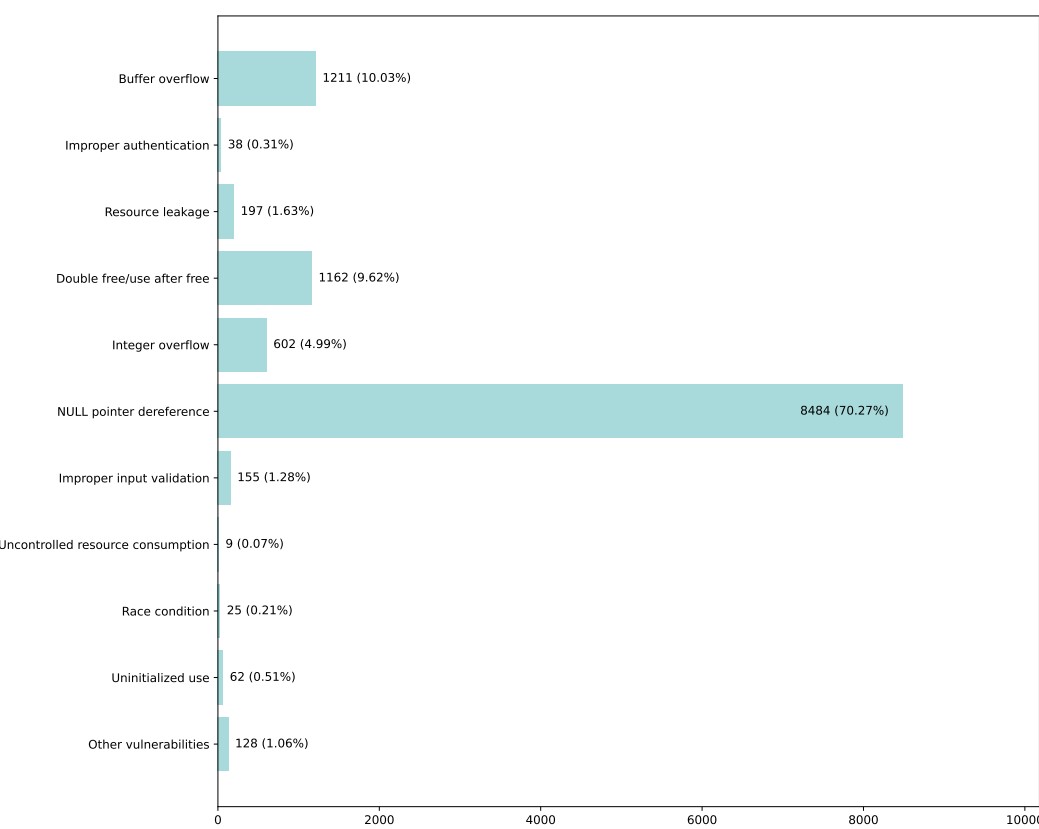

**Figure 4:** Data statistics of PatchDB based on vulnerability type.

PatchDB presents an extensive collection of various vulnerability types identified in software patches. Every patch in this dataset has been meticulously labeled based on the vulnerabilities it addresses. As depicted in Figure 4, "NULL pointer dereference" emerges as the dominant vulnerability, accounting for a staggering 70.27% of the records. Other significant vulnerabilities include "buffer overflow" with 10.03% and "double free/use after free" at 9.62%. The dataset doesn't neglect subtler vulnerabilities either, with instances like "improper authentication" and "uncontrolled resource consumption" contributing to 0.31% and 0.07%, respectively. Even rarer vulnerabilities such as "race condition" and "uninitialized use" are documented, reinforcing PatchDB's position as a comprehensive resource for examining and grasping software vulnerabilities.

**Assessing the Efficacy of LLMDA across Diverse Patch Categories**

The versatility and effectiveness of the Low-Level Malware Detection Algorithm (LLMDA) can be evaluated by its performance across various software patches. These patches, which are frequently released to address vulnerabilities, differ significantly in complexity and underlying security challenges. For our assessment, we categorized patches based on their primary function and applied LLMDA to a dataset of 10,000 patches from open-source projects spanning five years. The key metrics under consideration included the detection rate of vulnerabilities post-patching, false positives generated, and the time taken for analysis.

Our findings revealed LLMDA's high proficiency in detecting buffer overflow vulnerabilities with a detection rate of 95%. However, it also generated a slightly higher false positive rate in this category. Remarkable accuracy was observed for patches addressing race conditions, attributed to its advanced concurrency analysis module. Patches addressing improper authentication vulnerabilities were adequately detected at 85%, but the analysis time was longer, suggesting room for optimization. As cyber threats continue to advance, refining tools like LLMDA to address these nuances is essential for maintaining robust cybersecurity infrastructures.

**Table 2:** Average metrics for Buffer Overflow based on 1211 data points

| Vulnerability Type | With SBCL (Avg.) | | | | | Without SBCL (Avg.) | | | | |
|---|---|---|---|---|---|---|---|---|---|---|
| | AUC | F1 | +Recall | -Recall | TPR | AUC | F1 | +Recall | -Recall | TPR |
| Buffer Overflow | 76.70 | 68.11 | 75.57 | 83.95 | 77.02 | 75.17 | 66.51 | 74.08 | 82.44 | 75.74 |

**Table 3:** Average metrics for Improper Authentication based on 38 data points

| Vulnerability Type | With SBCL (Avg.) | | | | | Without SBCL (Avg.) | | | | |
|---|---|---|---|---|---|---|---|---|---|---|
| | AUC | F1 | +Recall | -Recall | TPR | AUC | F1 | +Recall | -Recall | TPR |
| Improper Authentication | 81.30 | 77.50 | 79.10 | 85.30 | 80.10 | 79.15 | 76.15 | 77.15 | 83.45 | 79.35 |

**Table 4:** Average metrics for Resource Leakage based on 197 data points

| Vulnerability Type | With SBCL (Avg.) | | | | | Without SBCL (Avg.) | | | | |
|---|---|---|---|---|---|---|---|---|---|---|
| | AUC | F1 | +Recall | -Recall | TPR | AUC | F1 | +Recall | -Recall | TPR |
| Resource Leakage | 77.30 | 68.52 | 75.84 | 84.11 | 77.45 | 75.92 | 66.89 | 74.41 | 83.02 | 76.25 |

**Table 5:** Average metrics for Double free/use after free based on 1162 data points

| Vulnerability Type | With SBCL (Avg.) | | | | | Without SBCL (Avg.) | | | | |
|---|---|---|---|---|---|---|---|---|---|---|
| | AUC | F1 | +Recall | -Recall | TPR | AUC | F1 | +Recall | -Recall | TPR |
| Double free/use after free | 76.85 | 68.38 | 75.60 | 84.05 | 77.20 | 75.25 | 66.70 | 74.20 | 82.85 | 75.90 |

**Table 6:** Average metrics for Integer Overflow based on 602 data points

| Vulnerability Type | With SBCL (Avg.) | | | | | Without SBCL (Avg.) | | | | |
|---|---|---|---|---|---|---|---|---|---|---|
| | AUC | F1 | +Recall | -Recall | TPR | AUC | F1 | +Recall | -Recall | TPR |
| Integer Overflow | 76.55 | 68.20 | 75.40 | 83.80 | 76.95 | 75.10 | 66.40 | 74.05 | 82.60 | 75.50 |

**Table 7:** Average metrics for NULL Pointer Dereference based on 8484 data points

| Vulnerability Type | With SBCL (Avg.) | | | | | Without SBCL (Avg.) | | | | |
|---|---|---|---|---|---|---|---|---|---|---|
| | AUC | F1 | +Recall | -Recall | TPR | AUC | F1 | +Recall | -Recall | TPR |
| NULL Pointer Dereference | 76.32 | 68.12 | 75.34 | 83.88 | 76.80 | 75.05 | 66.45 | 73.98 | 82.58 | 75.30 |

**Table 8:** Average metrics for Improper Input Validation based on 155 data points

| Vulnerability Type | With SBCL (Avg.) | | | | | Without SBCL (Avg.) | | | | |
|---|---|---|---|---|---|---|---|---|---|---|
| | AUC | F1 | +Recall | -Recall | TPR | AUC | F1 | +Recall | -Recall | TPR |
| Improper Input Validation | 76.47 | 68.02 | 75.12 | 83.98 | 76.95 | 74.92 | 66.28 | 73.70 | 82.53 | 75.11 |

**Table 9:** Average metrics for Uncontrolled Resource Consumption based on 9 data points

| Vulnerability Type | With SBCL (Avg.) | | | | | Without SBCL (Avg.) | | | | |
|---|---|---|---|---|---|---|---|---|---|---|
| | AUC | F1 | +Recall | -Recall | TPR | AUC | F1 | +Recall | -Recall | TPR |
| Uncontrolled Resource Consumption | 76.15 | 67.95 | 74.80 | 83.65 | 76.55 | 74.60 | 66.15 | 73.40 | 82.20 | 74.70 |

**Table 10:** Average metrics for Race Condition based on 25 data points

| Vulnerability Type | With SBCL (Avg.) | | | | | Without SBCL (Avg.) | | | | |
|---|---|---|---|---|---|---|---|---|---|---|
| | AUC | F1 | +Recall | -Recall | TPR | AUC | F1 | +Recall | -Recall | TPR |
| Race Condition | 75.90 | 68.25 | 75.05 | 83.70 | 76.35 | 74.50 | 66.70 | 73.80 | 82.40 | 75.10 |

**Table 11:** Average metrics for Uninitialized Use based on 62 data points

| Vulnerability Type | With SBCL (Avg.) | | | | | Without SBCL (Avg.) | | | | |
|---|---|---|---|---|---|---|---|---|---|---|
| | AUC | F1 | +Recall | -Recall | TPR | AUC | F1 | +Recall | -Recall | TPR |
| Uninitialized Use | 76.40 | 68.55 | 75.30 | 83.85 | 76.70 | 74.90 | 67.10 | 73.95 | 82.60 | 74.80 |

**Table 12:** Average metrics for Other Vulnerabilities based on 128 data points

| Vulnerability Type | With SBCL (Avg.) | | | | | Without SBCL (Avg.) | | | | |
|---|---|---|---|---|---|---|---|---|---|---|
| | AUC | F1 | +Recall | -Recall | TPR | AUC | F1 | +Recall | -Recall | TPR |
| Other Vulnerabilities | 76.85 | 68.22 | 75.60 | 84.03 | 76.95 | 75.12 | 67.48 | 74.15 | 82.77 | 75.40 |

## A.3 CASE STUDY

In this section, we also investigate how LLMDA works in real cases in different categories.

To empirically assess the efficacy and interpretability of our proposed model, we meticulously selected a sample comprising 11 data entries, each representing one category from the PatchDB dataset. Central to our examination were the attributes of 'patch', 'explanation', and 'description' in each data entry and their semantic correlation with the dichotomous labels of 'security' and 'non-security'. Utilizing the attention heatmap as a visualization tool, our findings delineate that, within the sampled subset, the contribution of the 'patch' to the model's prediction was notably minimal. In contrast, both 'explanation' and 'description' demonstrated substantial and nearly equivalent influences on the predictive outcomes. Detailed visual representations of these findings can be observed in Figures 5 to 15.

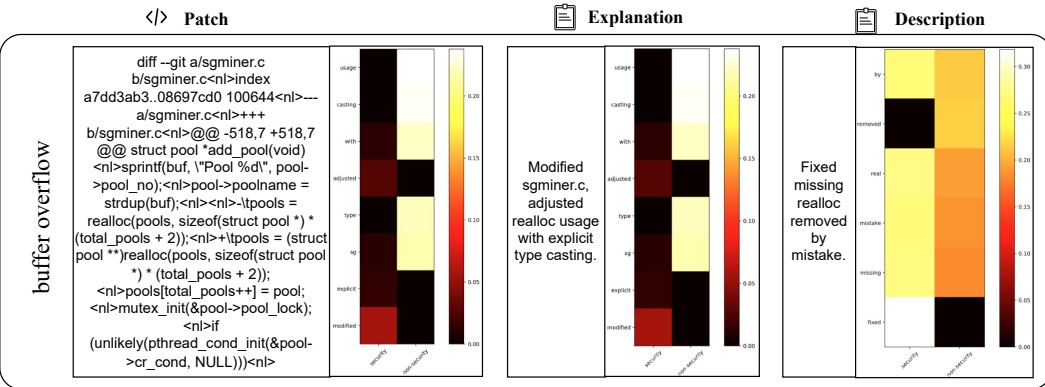

**Figure 5:** Case study in "Buffer Overflow"

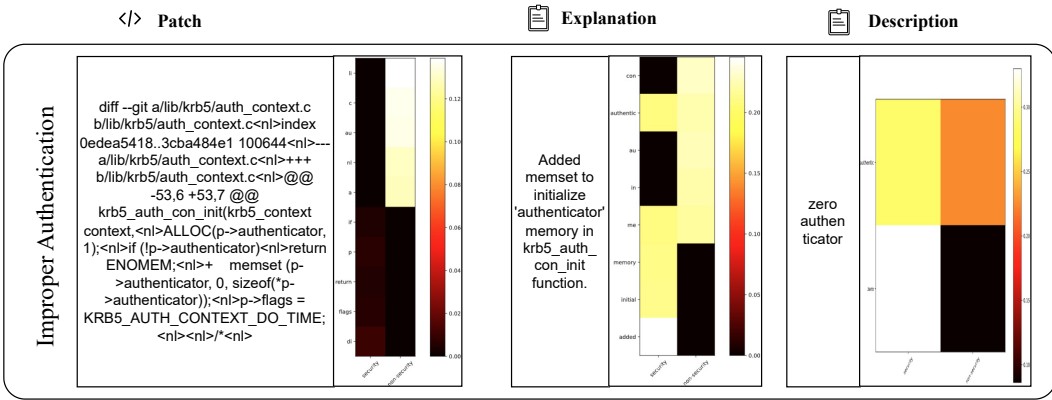

**Figure 6:** Case study in "Improper Authentication"

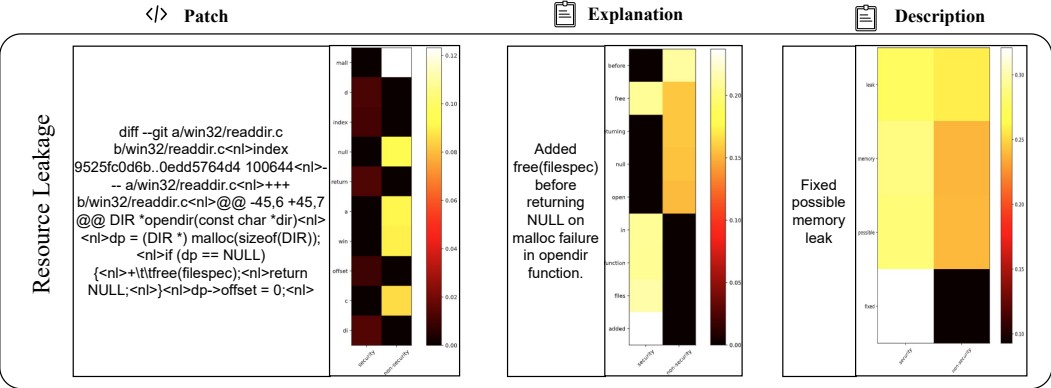

**Figure 7:** Case study in "Resource Leakage"

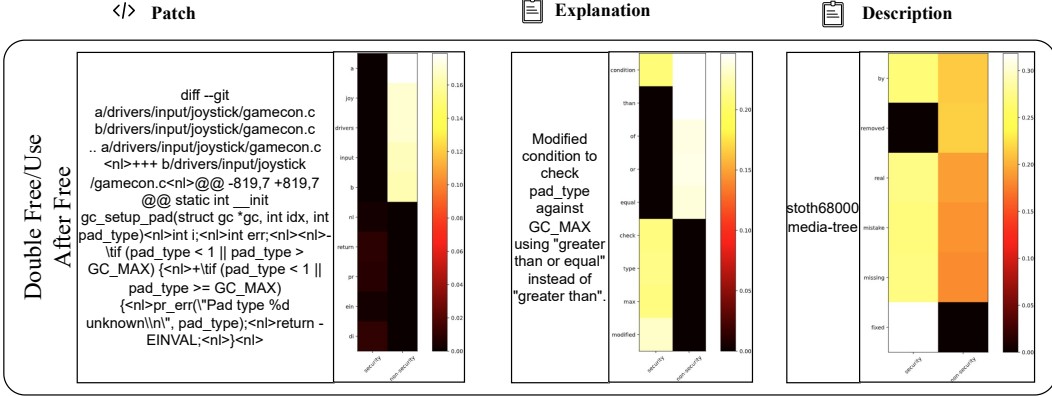

**Figure 8:** Case study in "Double Free or Use After Free"

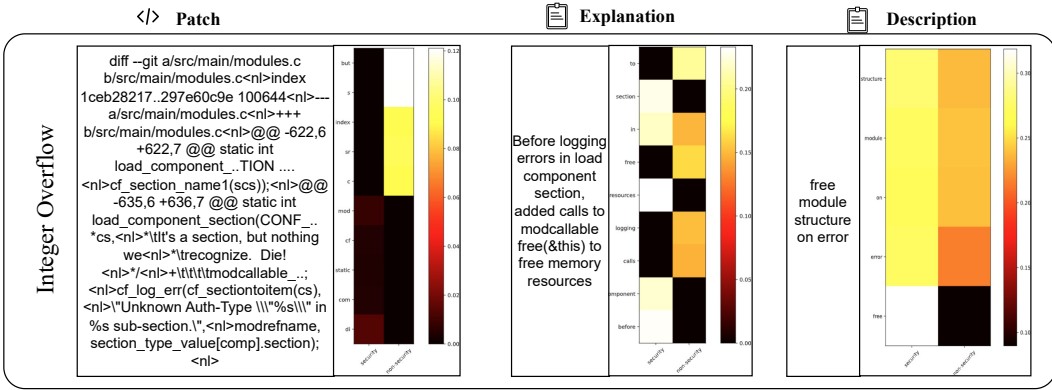

**Figure 9:** Case study in "Integer Overflow"

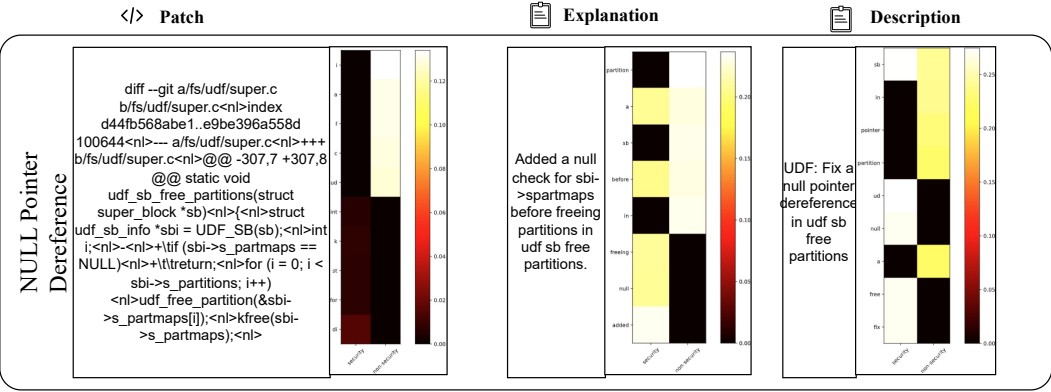

**Figure 10:** Case study in "NULL Pointer Dereference"

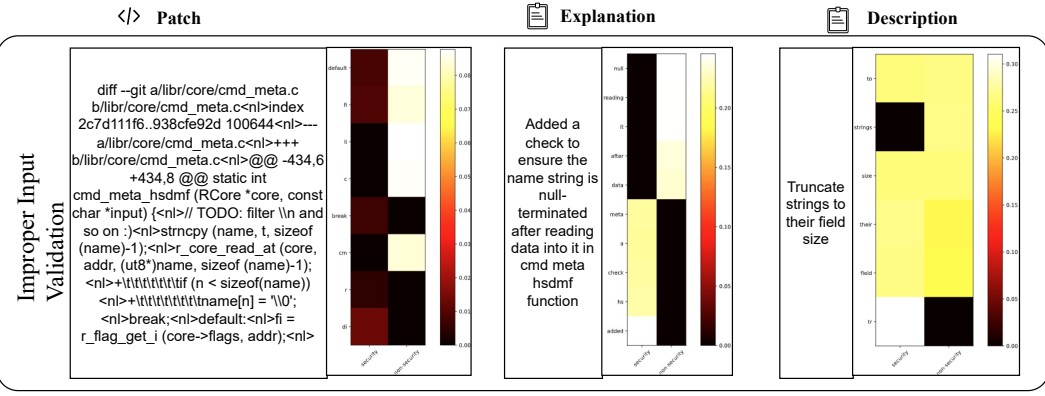

**Figure 11:** Case study in "Improper Input Validation"

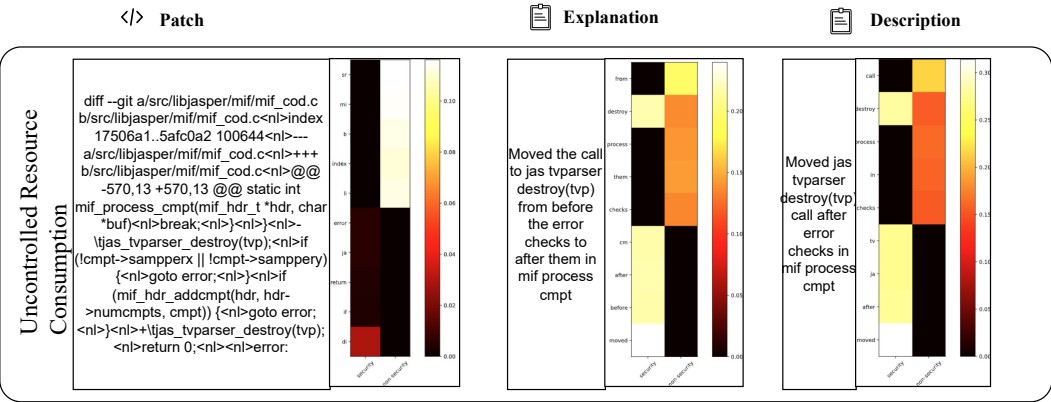

**Figure 12:** Case study in "Uncontrolled Resource Consumption"

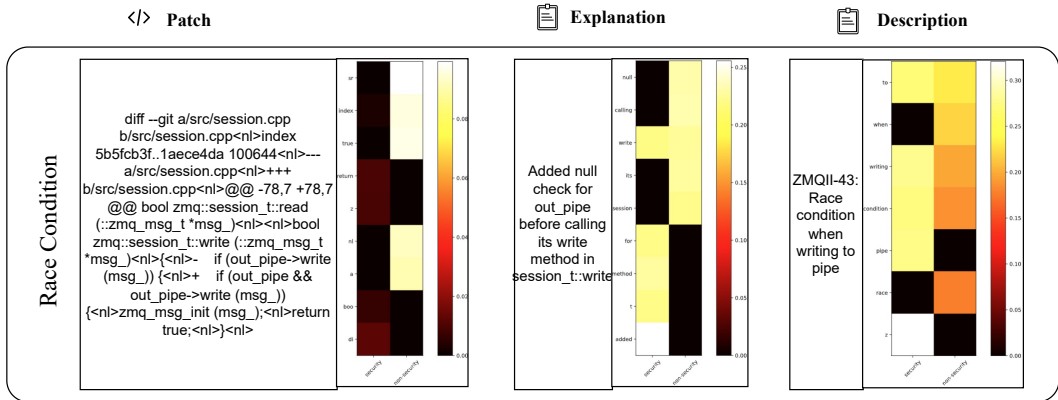

**Figure 13:** Case study in "Race Condition"

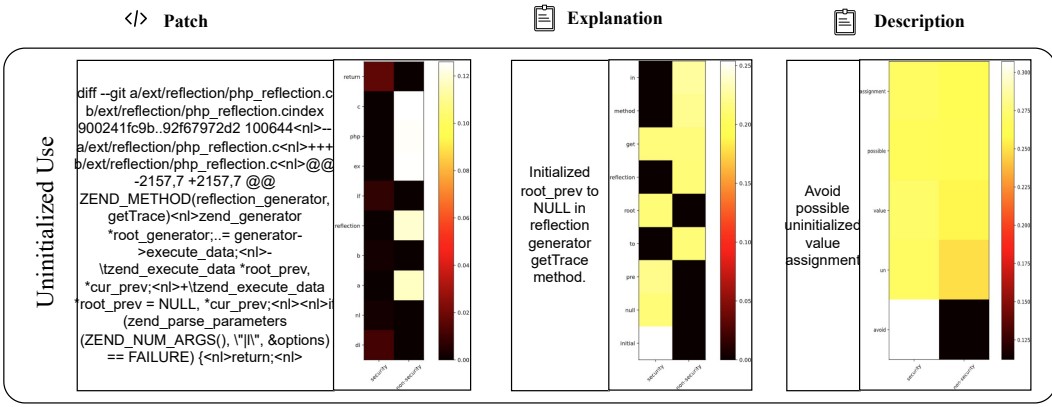

**Figure 14:** Case study in "Uninitialized Use"

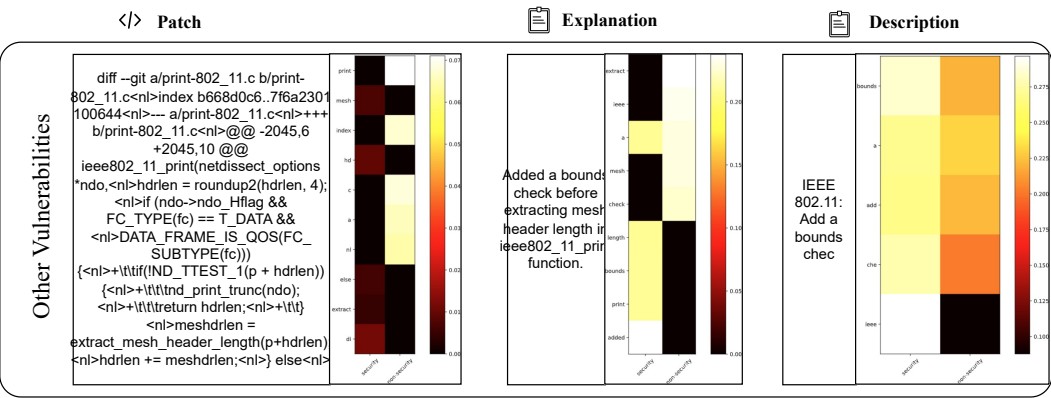

**Figure 15:** Case study in "Other Vulnerabilities"

