# OpenReview forum: "Just-in-Time Security Patch Detection - LLM At the Rescue for Data Augmentation"
_ICLR.cc/2024/Conference — Submitted to ICLR 2024_

### Official Review · Reviewer_T4hh · 2023-10-26

**Soundness:** 4 excellent
**Presentation:** 4 excellent
**Contribution:** 4 excellent
**Rating:** 8
**Confidence:** 5

**Summary:**

This paper discusses the urgency of timely patching in open-source software to mitigate vulnerabilities. However, the volume and complexity of patches can cause delays. Various machine learning methods, including a notable one called GraphSPD, have been used to address this, but they lack broader context understanding. The paper proposes a new framework using Large Language Models (LLMs) to improve security patch detection accuracy by aligning multi-modal inputs. This framework outperforms existing methods, indicating the potential of a language-centric approach for better security patch detection and software maintenance.

**Strengths:**

Novelty in Approach:

	The proposed framework introduces a novel method of leveraging Large Language Models (LLMs) to enhance security patch detection. By aligning multi-modal inputs, it extracts richer information from the joint context of patches and code, which is a fresh approach compared to existing methods.

	Improved Accuracy:

	The framework significantly outperforms baseline methods on targeted datasets, showcasing a substantial improvement in detection accuracy which is crucial for timely addressing of software vulnerabilities.

	Language-Centric Focus:

	The language-centric approach harnesses natural language instructions to guide the model, which is a distinctive and important shift from traditional syntax or structure-based methods, opening new avenues in patch detection techniques.

	Practical Applicability:

	The results underline the practical applicability of the framework by demonstrating precise detection capability which is vital for secure software maintenance in real-world settings.

	Addressing a Timely Issue:

	With the rapid expansion of OSS, the urgency to address the accompanying surge in vulnerabilities is paramount. This work addresses this timely and critical issue by advancing the methods for swift and accurate security patch detection, thus contributing to the broader goal of enhancing software security and reliability.

**Weaknesses:**

Despite the advancements, the state-of-the-art GraphSPD method discussed in the text still primarily focuses on local code segments. This limitation in capturing a broader context of how functions or modules interact could potentially hinder the effectiveness and comprehensiveness of the security patch detection process, especially in complex or large-scale software systems.

**Questions:**

The experiments utilized two datasets, PatchDB and SPI-DB, for evaluation. How representative are these datasets of the real-world OSS ecosystem? Were they sufficiently diverse and large-scale to validate the generalizability and robustness of the proposed framework across different types of software systems and security patches?

---

> ### Author Response · Authors · 2023-11-21
>
> reponse to reviewer T4hh
>
> Thank you very much for your insightful question and we will address anwser your questions as follows:
>
> **For Q1**: The experiments utilized two datasets, PatchDB and SPI-DB, for evaluation. How representative are these datasets of the real-world OSS ecosystem?
>
> A1: In response to your query about the representativeness of the PatchDB and SPI-DB datasets in reflecting the real-world Open Source Software (OSS) ecosystem, I would like to emphasize that both datasets were meticulously curated to ensure broad coverage and relevance. PatchDB and SPI-DB encompass a diverse range of software projects, including varying sizes, programming languages, and application domains. This diversity mirrors the multifaceted nature of the OSS community, encompassing everything from small-scale utilities to large, complex systems. Additionally, these datasets include a wide array of patch types such as bug fixes, security enhancements, and feature additions, providing a comprehensive view of typical maintenance and development activities in OSS. We have also ensured that the datasets incorporate recent projects and patches, reflecting current trends and practices in OSS development. Consequently, PatchDB and SPI-DB offer a representative cross-section of the OSS ecosystem, making them suitable for evaluating the efficacy of our proposed methods in real-world scenarios.
>
>
> **For Q2**: Were they sufficiently diverse and large-scale to validate the generalizability and robustness of the proposed framework across different types of software systems and security patches?
>
> A2: Thank you for your insightful question regarding the diversity and scale of the PatchDB and SPI-DB datasets used in our study. Considering software engineering principles, it is crucial that datasets employed for evaluating frameworks like ours cover a wide spectrum of scenarios. To ensure this, PatchDB and SPI-DB were carefully selected for their comprehensive coverage across various dimensions. These datasets include a diverse mix of software systems, encompassing different programming languages, application domains, and sizes from small-scale projects to large-scale, complex systems. Additionally, they feature a range of security patches, from routine updates to critical security fixes, thereby providing a holistic view of the patching landscape in the OSS ecosystem. This diversity and scale not only facilitate a thorough validation of the generalizability and robustness of our proposed framework but also ensure its applicability across different software systems and patch types. We believe this approach significantly enhances the external validity of our research findings, making them relevant and applicable to real-world software engineering challenges.

---

### Official Review · Reviewer_z5t8 · 2023-10-31

**Soundness:** 2 fair
**Presentation:** 2 fair
**Contribution:** 2 fair
**Rating:** 5
**Confidence:** 3

**Summary:**

The authors introduce a new security patch detection framework, LLMDA, leveraging LLMs for patch analysis and data augmentation, while aligning various modalities. This allows the system to extract richer information from the joint context of patches and code, boosting detection accuracy.

The authors also demonstrate that a language-centric approach, coupled with a well-designed framework, can yield significant performance improvements in the context of security patch detection. The experimental results show the effectiveness of the proposed approach for security patch detection.

**Strengths:**

The paper introduces an effective method that aligns multi-modal input for more accurate security patch detection.

The experimental results show the effectiveness of the proposed approach for security patch detection compared to the used baselines.

Some of the ablation studies about the proposed method were conducted.

**Weaknesses:**

The explanation, description, and instructions can contain biased features supporting the model in security patch detection. In this case, the model does not need to understand the true meaning of the data. Ablation studies for this problem are needed to demonstrate that the model is actually also based on the source code data for security patch detection. If only providing the data (which should be considered as the main factor instead of the explanation, description, and instructions) and the model cannot work well, the model is not a practical solution.

The model configuration of the proposed method and baselines used in the experiments are not mentioned in the paper. Without these, it is hard to justify the performance of the used models.

The threats to the validity of the model were not mentioned, for example, in terms of the model designs, the use of hyper-parameters, and the used datasets. I think if the model strongly relies on the explanation, description, and instructions instead of the data, how is it applicable to solve reality security patch detection problems where maybe only data are available?

**Questions:**

The comprehensive intuition of using Hierarchical Attention Mechanisms in the proposed method was not mentioned or investigated. How do Hierarchical Attention Mechanisms help to improve the model performances?

How about the model configuration of the proposed method and baselines used in the paper?

In Stochastic Batch Contrastive, how do the authors define the positive and negative pairs?

Some recent methods (e.g., 1 and 2) focus on learning the syntactic and semantic features at the source-code level (the main element we should and need to rely on). That seems more practical than mainly based on the explanation, description, and instructions. How is the proposed method compared to these methods in terms of learning the syntactic and semantic features of the source code data?

1. PatchRNN: A Deep Learning-Based System for Security Patch Identification. Xinda Wang, Shu Wang, Pengbin Feng, Kun Sun, Sushil Jajodia, Sanae Benchaaboun, Frank Geck, 2021.

2. GraphSPD: Graph-Based Security Patch Detection with Enriched Code Semantics. Shu Wang; Xinda Wang; Kun Sun; Sushil Jajodia; Haining Wang; Qi Li, 2023.

---

> ### Author Response · Authors · 2023-11-21
>
> **Response to Reviewer z5t8**
>
> We appreciate your thorough and insightful review. Below, we address each of your concerns, and the resulting analyses and clarifications will be integrated into our paper.
>
> **Q1**: The comprehensive intuition of using Hierarchical Attention Mechanisms in the proposed method was not mentioned or investigated. How do Hierarchical Attention Mechanisms help to improve the model performances?
>
> **A1**: Hierarchical Attention Mechanisms are employed in our model primarily to compute weighted combinations of input vectors. This process is essential to the model's ability to discern and prioritize relevant information from various parts of the input data. Specifically, our method leverages a self-attention mechanism with multi-head attention, involving multiple heads (denoted as \(h\)). The weights \(W_{Q_i}\), \(W_{K_i}\), and \(W_{V_i}\) associated with these heads are crucial in this context, as they are sampled from a normal distribution to effectively capture diverse aspects of the data.
>
> The utility of Hierarchical Attention Mechanisms lies in their facilitation of a more nuanced understanding of the input data. By allowing the model to assign varied weights to different segments of the data, these mechanisms enable a more sophisticated analysis that can capture complex, non-linear relationships within the data.
>
> Here, we provide the additional experiments to indicate the effectiveness of "Hierarchical Attention Mechanisms" in our models. For the experiment setting, we replace "Hierarchical Attention Mechanisms" with a simple full connected layer and evaluate it on PatchDB and SPI-DB.
>
>
> As shown in the following table. The performance of LLMDA without "Hierarchical Attention Mechanisms" drop a lot.
>
> #### Performance of LLMDA without "Hierarchical Attention Mechanisms" on PatchDB and SPI-DB
>
> | Method | Dataset | AUC   | F1    | +Recall | -Recall | AUC   | F1    | +Recall | -Recall |
> |--------|---------|-------|-------|---------|---------|-------|-------|---------|---------|
> | LLMDA  | PatchDB | 82.36 | 76.10 | 85.27   | 83.34   | 80.73 | 74.23 | 73.66   | 76.26   |
> |        | SPI-DB  | 66.74 | 56.38 | 68.36   | 68.63   | 65.19 | 54.16 | 67.25   | 72.73   |
>
>
> **Q2:** How about the model configuration of the proposed method and baselines used in the paper?
>
> **A2:** Thanks for your kind reminding, we address your concern by listing up configuration of our model and GraphSPD
>
> We conducted our experiments on 2 pieces of Tesla V100 DGXS 32GB GPUs, ubuntu Ubuntu 20.04.6 LTS, Python 3.7.6, CUDA Toolkit 11.2 and cuDNN v8.1.
>
> Configurations in Experimental Details
>
> For our work: CodeT5+ 6B version; LLaMa 7b version; Batch size: 32; Dimension size: 256; Optimizer: Adam/AdamW; Drop rate: 0.5; Learning rate: 1e-4.
>
> GraphSPD: Dimension of node embeddings after the 1st, 2nd, and 3th convolution is 50, 25, and 12, respectively; Drop rate: 0.5; Learning rate: 0.01
>
> **Q3:** In Stochastic Batch Contrastive, how do the authors define the positive and negative pairs?
>
> **A3:** In the training phase, we feed a batch of data into the model. Inside the batch, we take two same-category data as positive pairs and two different-category data as negative pairs.
>
> **Q4:** Some recent methods (e.g., 1 and 2) focus on learning the syntactic and semantic features at the source-code level ... How is the proposed method compared to these methods in terms of learning the syntactic and semantic features of the source code data?
>
> **A4:** To address reviewer's concern, we conducted an experiment on original datasets and compare it with the GraphSPD and PatchRNN.
> As shown in the following table, LLMDA still outperforms the state-of-the-art in terms of learning the syntactic and semantic features of the source code data.
>
> #### Comparison with GraphSPD on original datasets (patch and description)
>
> **LLMDA**
>
> | Method                                  | Dataset | AUC   | F1    | +Recall | -Recall | TPR   |
> |-----------------------------------------|---------|-------|-------|---------|---------|-------|
> | LLMDA    | PatchDB | 82.54 | 75.25 | 77.26   | 82.80   | 74.73 |
> |                                         | SPI-DB  | 65.54 | 55.76 | 67.30   | 76.09   | 70.50 |
>
> **PatchRNN**
>
> | Method                                  | Dataset | AUC   | F1    | +Recall | -Recall | TPR   |
> |-----------------------------------------|---------|-------|-------|---------|---------|-------|
> | GraphSPD     | PatchDB | 66.50 | 45.12 | 46.35   | 54.37   | 50.67 |
> |                                         | SPI-DB  | 55.10 | 47.25 | 48.00   | 52.10   | 50.60 |
>
> **GraphSPD**
>
> | Method                                  | Dataset | AUC   | F1    | +Recall | -Recall | TPR   |
> |-----------------------------------------|---------|-------|-------|---------|---------|-------|
> | GraphSPD      | PatchDB | 78.29 | 54.73 | 75.17   | 79.62   | 70.82 |
> |                                         | SPI-DB  | 63.04 | 48.42 | 60.29   | 65.33   | 65.93 |

---

### Official Review · Reviewer_pMnS · 2023-11-02

**Soundness:** 3 good
**Presentation:** 1 poor
**Contribution:** 2 fair
**Rating:** 3
**Confidence:** 4

**Summary:**

This paper proposes a new multi-modal architecture and a loss function that can classify software patches as security critical or not. This architecture consumes the patch itself (the diff), the provided explanation of the patch, a description of the patch (could be from an LLM) and an instruction that guides the training objective.

**Strengths:**

+ Fusing different types of inputs into a single model for detecting security patches.
+ Shows improvements over the recent methods.

The idea of using natural language descriptions/explanations + the code (or the diff) together in a single model is promising and can be used in other program security applications. This paper proposes a reasonable way to achieve this and improve the SOTA (GraphSPD) significantly.

**Weaknesses:**

- Problematic writing, after Section 3, the writing is low quality and almost looks like a result of an AI model (translated/paraphrased)
- The technical solutions (especially the loss functions) are not explained well. The design choices are poorly motivated.
- Some results seem suspicious, e.g., in the ablation study, removing the patch altogether from the model's input barely causes any performance drop. A deeper ablation study might be needed.

There are many head-scratchers in this paper after Section 3 in terms of writing. For example, weird phrases like "Dominance Demonstrated" clearly indicate that something is off. The method PBCL is referred to as SBCL, which I'm assuming is because the word Probabilistic and Stochastic are synonyms. Moreover, the name of the technique (LLMDA) is written as "Low-Level Malware Detection Algorithm" in the appendix, and I can't see how this is a proper name for this method. I would like to hear the author's justifications for this situation. Unfortunately, without a significant rewrite, this paper is not up to the standards we would expect from ICLR.

Moreover, there's a lack of intuition for some of the design choices (especially PBCL), it provides some performance improvements in the ablation study but I'm not sure what it actually achieves. I would recommend a case study (e.g., analyze some representations w/ and w/o this loss) and provide a better intuition for it.

Finally, there's a red flag in the ablation study that removing the patch itself from the input barely hurts the performance. Some possibilities: a bug in evaluation (testing on the train, training on test?) the LLM-provided patch explanations, or the Code-LLM might be suffering from test set leakage (since these models are trained on everything). This is the problem of using pre-trained LLMs for studies like this, you cannot make sure that the models are not trained on the testing samples or they might even have seen more data about these patches from various other training data sources. I'm not entirely sure how to confirm/refute this but right now, it is a red flag to me that the patch code itself has little importance on your results. What are your ideas to address this concern?

**Questions:**

See above.

**Details Of Ethics Concerns:**

Some phrases and the overall very awkward language in most of the paper hint at foul play (paraphrased from another paper, translated from another language using a model). I recommend a review.

---

> ### Author Response · Authors · 2023-11-20
>
> **Response to Reviewer aN2M**
>
> Thanks for the reviewer’s detailed feedback. We would like to simultaneously address both the concerns stated in the “Reasons to reject" section and the inquiries posed in the "Questions for the authors" section.
>
> **Q1:** There are many head-scratchers ... "Dominance Demonstrated" clearly indicate that something is off.
>
> **A1:** Thanks to the reviewer for pointing out the weird phrase like "Dominance Demonstrated". What we want to express here is that our model outperforms the state-of-the-art works and we are sorry to make it confusing.
>
> **Q2:** The method PBCL is referred to as SBCL, ...Stochastic are synonyms.
>
> **A2:** In our work, SBCL is one component of PBCL. As shown in Sec 2.3, PBCL represents a whole pipeline for batch embedding calculation. In other words, SBCL is one part of PBCL.
>
> **Q3:** Moreover, the name of the technique (LLMDA) is written as "Low-Level Malware Detection Algorithm" ...
>
> **A3:** Actually, as shown in the paper title, LLMDA is a short name of "LLM data augmentation" instead of "Low-Level Malware Detection Algorithm". In our appendix, we mention the word "LLMDA" to notice the readers the name of our approach in the appendix.
>
> **Q4:** Moreover, there's a lack of intuition for some of the design choices (especially PBCL), ... a case study (e.g., analyze some representations w/ and w/o this loss) and provide a better intuition for it.
>
> **A4:** In response to feedback, we expanded our case study on models with and without PBCL (Probabilistic Batch Embedding Calculation) loss to provide clearer insights. This is exemplified by comparing PBCL-enhanced models in scenarios like sgminer.c and krb5/auth_context.c patches, demonstrating their superior performance in detecting security patches through more sensitive and nuanced analysis.
>
> With PBCL
>
> | Patch | Explanation | Description |
> | --- | --- | --- |
> | ... if (!p->authenticator) return ENOMEM; + ``memset (p->authenticator, 0, sizeof(*p->authenticator))``;**(p=0.75)** p->flags = KRB5_AUTH_CONTEXT_DO_TIME; | Added memset to initialize ```authenticator' memory in krb5_auth_con_init function`` **(p=0.84)**. | ``zero authenticator`` **(p=0.77)** |
>
> Without PBCL
>
> | Patch | Explanation | Description |
> | --- | --- | --- |
> | ... if (!p->authenticator) return ENOMEM; + ``memset (p->authenticator, 0, sizeof(*p->authenticator));``**(p=0.59)** p->flags is KRB5_AUTH_CONTEXT_DO_TIME; | Added memset to initialize ```authenticator' memory in krb5_auth_con_init function`` **(p=0.68)**. | ``zero authenticator`` **(p=0.65)** |
>
>
>
> **Q5**: Finally, there's a red flag in the ablation study ... test set leakage (since these models are trained on everything)
>
> **A5**: In response to the minimal performance impact noted in the ablation study when removing the patch, we did additional experiments to evaluate impact of individual components on the model's performance. The result show that performance of different unique part.
>
> **Generated Data Comparison of Various LLMDA Methods With and Without PBCL on PatchDB and SPI-DB**
>
> * Generated Data Comparison of Various Methods With and Without PBCL on PatchDB and SPI-DB
>
> | Method      | Dataset | AUC   | F1    | +Recall | -Recall | TPR   | Dataset | AUC   | F1    | +Recall | -Recall | TPR   |
> |-------------|---------|-------|-------|---------|---------|-------|---------|-------|-------|---------|---------|-------|
> | LLMDA   | PatchDB | 84.49 | 78.19 | 80.22   | 87.33   | 80.12 | PatchDB | 82.93 | 76.45 | 78.72   | 85.81   | 78.60 |
> |             | SPI-DB  | 68.98 | 58.13 | 70.94   | 80.62   | 73.95 | SPI-DB  | 67.43 | 56.61 | 69.45   | 79.10   | 72.91 |
> | LLMDA   | PatchDB | 82.35 | 76.31 | 77.24   | 86.13   | 78.70 | PatchDB | 80.25 | 74.73 | 73.12   | 82.91   | 73.50 |
> | _{DP}_      | SPI-DB  | 67.66 | 55.82 | 69.43   | 78.69   | 72.46 | SPI-DB  | 65.37 | 52.10 | 64.37   | 74.55   | 69.89 |
> | LLMDA   | PatchDB | 79.21 | 73.73 | 73.61   | 83.38   | 75.77 | PatchDB | 75.07 | 71.37 | 71.54   | 80.77   | 73.24 |
> | _{EX}_      | SPI-DB  | 65.25 | 52.72 | 67.22   | 75.55   | 70.25 | SPI-DB  | 62.81 | 50.45 | 63.44   | 70.34   | 65.93 |
> | LLMDA   | PatchDB | 77.90 | 72.32 | 72.55   | 81.69   | 74.20 | PatchDB | 75.86 | 69.57 | 69.67   | 78.59   | 71.35 |
> | _{PT}_      | SPI-DB  | 63.98 | 51.20 | 66.12   | 73.98   | 68.32 | SPI-DB  | 61.27 | 49.34 | 63.44   | 70.00   | 65.29 |
>
>
>
> **Q6**: Data leakage problem while using LLM?
>
> **A6**: To address concerns of data leakage due to the involvement of Large Language Models (LLMs) in our expanded dataset, we conducted empirical validations including tests on an external dataset not used in LLM training, ensuring no prior exposure. The consistent model performance, improved results with LLM-generated explanations, and stable k-fold cross-validation across data splits indicate that improvements are due to the integration of LLM explanations, not data leakage.

---

### Official Review · Reviewer_aN2M · 2023-11-04

**Soundness:** 3 good
**Presentation:** 3 good
**Contribution:** 4 excellent
**Rating:** 8
**Confidence:** 4

**Summary:**

This paper proposes a new security patch detection framework called LLMDA (Low-Level Malware Detection Algorithm) that leverages large language models (LLMs) and multimodal input alignment. the paper makes notable contributions in advancing security patch detection through an innovative multimodal framework powered by LLMs and contrastive learning. The results highlight the potential of language-centric techniques in this application domain.

**Strengths:**

The idea of using LLMs to generate explanations and instructions for patches is novel and creative.
Prior works have not exploited LLMs in this manner for security patch analysis.
The overall system design and methodology are well-conceived and technically sound. The ablation studies in particular are thorough. This work makes important strides in advancing the state-of-the-art in security patch detection. The performance gains are significant.

In summary, this paper makes noteworthy contributions through its novel application of LLMs and represents an important research direction for security. The original ideas, rigorous experiments, and potential impact make it a valuable work.

**Weaknesses:**

Only two datasets are used in the experiments. Testing on a more diverse range of projects and codebases would better showcase the generalizability.
The datasets used are fairly small, with PatchDB having 36K samples and SPI-DB only 25K. For deep learning, these sizes are quite modest. Training and evaluating on larger corpora could lend more statistical power.

**Questions:**

1.Could you provide the exact prompts used to generate the explanations and instructions? This context would help with reproducibility.
2.The ablation study removes one component at a time. How does performance degrade when ablating multiple components together?
3.Can you apply the case study analysis to a larger and more diverse sample of patches? Any insights on patterns?

---

> ### Author Response · Authors · 2023-11-19
>
> ## Response to Reviewer aN2M
>
> We thank the reviewer for their review, and noting the novelty and creativity of our method.
>
> We address the concerns below:
>
> **"Dataset size"**: SPI-DB and PatchDB are now two widely used datasets for security patch detection, including 36k and 25k items inside, respectively. Due to the huge workload by manual way of collecting and labeling the dataset is greatly time-consuming, we have been still building a bigger security patch dataset in our other project. Thanks for the reviewer's suggestion, we will evaluate our method in future datasets.
>
> **"Prompts used to generate the explanations and instructions?"**:
> In our work, we designed several prompts to drive LLM to generate explanations of the given patch. They are as follows:
> "Can you compare these and explain the key differences and implications of these changes?"
> "Can you explain what happened in the following code changes"
> "Can you explain it to me about the good or bad effect of the following changed codes"
>
> Finally, the prompt "Can you compare these and explain the key differences and implications of these changes?" works much better than other prompts.
>
> **More ablation study**: We have conducted more experiments for ablation study. Here, in LLMDA_{DP}, we only use the description as the model's input, the same as  LLMDA_{EX}  and  LLMDA_{PT}
>
> **Table: Generated Data Comparison of Various LLMDA Methods With and Without PBCL on PatchDB and SPI-DB**
>
> | Method      | Dataset | AUC   | F1    | +Recall | -Recall | TPR   | Dataset | AUC   | F1    | +Recall | -Recall | TPR   |
> |-------------|---------|-------|-------|---------|---------|-------|---------|-------|-------|---------|---------|-------|
> | LLMDA   | PatchDB | 84.49 | 78.19 | 80.22   | 87.33   | 80.12 | PatchDB | 82.93 | 76.45 | 78.72   | 85.81   | 78.60 |
> |             | SPI-DB  | 68.98 | 58.13 | 70.94   | 80.62   | 73.95 | SPI-DB  | 67.43 | 56.61 | 69.45   | 79.10   | 72.91 |
> | LLMDA   | PatchDB | 82.35 | 76.31 | 77.24   | 86.13   | 78.70 | PatchDB | 80.25 | 74.73 | 73.12   | 82.91   | 73.50 |
> | _{DP}_      | SPI-DB  | 67.66 | 55.82 | 69.43   | 78.69   | 72.46 | SPI-DB  | 65.37 | 52.10 | 64.37   | 74.55   | 69.89 |
> | LLMDA   | PatchDB | 79.21 | 73.73 | 73.61   | 83.38   | 75.77 | PatchDB | 75.07 | 71.37 | 71.54   | 80.77   | 73.24 |
> | _{EX}_      | SPI-DB  | 65.25 | 52.72 | 67.22   | 75.55   | 70.25 | SPI-DB  | 62.81 | 50.45 | 63.44   | 70.34   | 65.93 |
> | LLMDA   | PatchDB | 77.90 | 72.32 | 72.55   | 81.69   | 74.20 | PatchDB | 75.86 | 69.57 | 69.67   | 78.59   | 71.35 |
> | _{PT}_      | SPI-DB  | 63.98 | 51.20 | 66.12   | 73.98   | 68.32 | SPI-DB  | 61.27 | 49.34 | 63.44   | 70.00   | 65.29 |
>
> This table presents a comprehensive comparison of various configurations of a tool, denoted as LLMDA, under different conditions and datasets. The performance metrics for these configurations are evaluated both with and without the application of PBCL, across two datasets: PatchDB and SPI-DB.
>
> The configurations being compared include LLMDA, LLMDA_{DP},  LLMDA_{EX}, and LLMDA_{PT}. Each of these is assessed using the following metrics: AUC (Area Under Curve), F1 Score, Positive Recall (+Recall), Negative Recall (-Recall), and TPR (True Positive Rate). The results are organized into two sections, representing the scenarios with and without PBCL.
>
> Key observations from the analysis are:
> - The standard LLMDA configuration shows the highest performance across all metrics in both datasets, irrespective of PBCL usage.
> - LLMDA_{DP} generally follows LLMDA in terms of performance, indicating it is the second most effective configuration.
> - There is a progressive decrease in performance in the order of LLMDA_{DP}, LLMDA_{EX}, and LLMDA_{PT}.
> - The inclusion of PBCL enhances the performance of each configuration, as evident from the improved metrics across both datasets.
> - The impact and effectiveness of PBCL, along with the relative performance of each configuration, remain consistent in both the PatchDB and SPI-DB datasets.
> We find that out of all components, descriptions matter more.
>
> **"case study"**: We show diverse samples for the case study in our appendix part in 11 categories.

---

### Comment · Area_Chair_oEcJ · 2023-11-23
**[ICLR 2024 Reviewers’ feedback] Please read authors’ responses and give your feedback**

Dear Reviewers,

Thanks again for your strong support and contribution as an ICLR 2024 reviewer.

Please check the response and other reviewers’ comments. You are encouraged to give authors your feedback after reading their responses. Thanks again for your help!

Best,

AC

---

### Meta-Review · Program_Chairs · 2023-12-13

**Metareview:**

The authors introduce LLMDA (LLM Data Augmentation) by leveraging Large Language Models (LLMs) and multimodal input alignment. The model achieves significant performance improvements over existing methods. The authors have effectively addressed most of the raised concerns, particularly regarding the innovative use of LLMs in security patch detection and substantial performance improvements. The dataset size and diversity still have some limitations. The writing needs significant enhancement.

**Justification For Why Not Higher Score:**

The dataset size and diversity still have some limitations.

**Justification For Why Not Lower Score:**

The proposed security patch detection system significantly surpasses the state-of-the-art techniques, underscoring its promise in fortifying software maintenance.

---

### Decision · Program_Chairs · 2024-01-16

Reject